# Pushing the Limits of Prenatal Ultrasound: A Case of Dorsal Dermal Sinus Associated with an Overt Arnold–Chiari Malformation and a 3q Duplication

Olivier Leroij [1], Lennart Van der Veeken [2,*], Bettina Blaumeiser [3] and Katrien Janssens [3]

1    Faculty of Medicine, University of Antwerp, 2610 Wilrijk, Belgium; olivier.leroij@uza.be
2    Department of Obstetrics and Gynaecology, University Hospital Antwerp, 2650 Edegem, Belgium
3    Department of Medical Genetics, University Hospital and University of Antwerp, 2650 Edegem, Belgium; bettina.blaumeiser@uantwerpen.be (B.B.); katrien.janssens@uantwerpen.be (K.J.)
*    Correspondence: lennart.vanderveeken@uza.be

**Abstract:** We present a case of a fetus with cranial abnormalities typical of open spina bifida but with an intact spine shown on both ultrasound and fetal MRI. Expert ultrasound examination revealed a very small tract between the spine and the skin, and a postmortem examination confirmed the diagnosis of a dorsal dermal sinus. Genetic analysis found a mosaic 3q23q27 duplication in the form of a marker chromosome. This case emphasizes that meticulous prenatal ultrasound examination has the potential to diagnose even closed subtypes of neural tube defects. Furthermore, with cerebral anomalies suggesting a spina bifida, other imaging techniques together with genetic tests and measurement of alpha-fetoprotein in the amniotic fluid should be performed.

**Keywords:** dorsal dermal sinus; Arnold–Chiari anomaly; 3q23q27 duplication; mosaic; marker chromosome

## 1. Introduction

Neural tube defects are the most common central nervous system anomaly. Two major types are distinguished: the open spina bifida aperta and closed spina bifida occulta. Open spina bifida lesions are usually diagnosed prenatally by means of an ultrasound based on the characteristic brain abnormalities: the lemon-sign, which reflects the abnormal skull shape, and the banana-sign, reflecting the abnormal cerebellar shape. Furthermore, in open spina bifida, a dorsal sac (myelomeningocele) or missing processi spinosi can easily be detected on ultrasound, which confirms the diagnosis. Closed spina bifida lesions are usually more difficult to diagnose based on prenatal ultrasound because of the absence of cranial abnormalities. The diagnosis can still be made if the lesion is characterized by a skin-covered sac (meningocele). However, certain subtypes of closed spina bifida are more subtle and are, therefore, often missed during prenatal ultrasound. Dorsal dermal sinuses are one of the subtypes of closed spina bifida lesions. These lesions are characterised by an epithelial-lined tract that connects the skin surface with the intracanalicular space. Due to the absence of secondary cranial abnormalities, as found in open spina bifida lesions, and the absence of cysts or masses at the spine, these lesions are usually missed during prenatal ultrasound screening. Alfa-fetoprotein levels in amniotic fluid can be measured to help to find the diagnosis. If there is leakage of spinal fluid, these levels will rise. This technique is presently rarely performed; however, in difficult or atypical cases, it can still assist in confirming the diagnosis. We present a case of a fetus with cranial abnormalities typical of open spina bifida but with an intact spine on ultrasound and fetal MRI. Here, we want to demonstrate that even in closed spina bifida, cranial abnormalities can be present. Furthermore, when cranial abnormalities are detected during prenatal ultrasound but the spine appears to be intact, alfa-fetoprotein remains a powerful tool to help diagnose prenatal atypical spina bifida cases.

## 2. Case Presentation

A 27-year-old primigravida was referred to the outpatient clinic of our hospital at a gestational age (GA) of 17 + 3 weeks following detection of a fetal unilateral hydronephrosis by the peripheral gynecologist. Ultrasonography demonstrated a duplex collecting system in the left kidney with hydronephrosis of the cranial pole, moderate dilatation of the upper pole ureter, and mild dilatation of the lower pole ureter. The cerebellum at that time showed a minimal posterior curving (Figure 1), but no distinct banana-sign. Inspection of the spine was inconspicuous. The patient was reevaluated in our center at 24 weeks GA. The hydronephrosis of the upper pole of the left kidney remained stable. However, the cerebellum presented a banana-shape with herniation of the hindbrain onto C2 and minimal lemon-sign of the skull (Figure 2). The lateral ventricles were not dilated. Ultrasound examination of the spine revealed a closed skin except for a minimal fistula in the sacro-coccygeal region (Figures 3–5). Furthermore, an abnormal gap between the second and third digit was seen on both hands as well as a syndactyly between the first and second toe. Cardiac sonography was normal at both 17- and 24-weeks GA, with a normal four-chamber view, outflow tracts, and three-vessel view. An amniocentesis was performed to rule out genetic anomalies and to measure alfa 1-fetoprotein. Additionally, a fetal MRI was carried out mainly to assess the fetal spine, as the cerebral findings all indicated the presence of an open neural tube defect. MRI analysis confirmed the hydroureteronephrosis as well as the lemon-shaped skull, Arnold–Chiari malformation 11mm below the foramen magnum, absent cerebrospinal fluid around the cerebellum, and possible mild tethering of the cord. However, MRI failed to demonstrate an open spina bifida or any skin defects. The Alfa 1-fetoprotein was elevated, further reinforcing the assumption of a classic spina bifida. QF-PCR showed no aneuploidy for chromosomes 13, 18, 21, X, or Y, but SNP array demonstrated a terminal multiplication of approximately 56 Mb of the long arm (3q23qter) of chromosome 3: arr 3q23q29(141903905-197845233)x2~4. The nature of the multiplication could not be deduced. Interphase FISH with probes in chromosomal regions 3q26 and 3q27 showed 4 copies of both chromosomal bands in approximately 50% of the interphases. Metaphase FISH proved that the 2 extra copies were located on a marker chromosome. Chromosomal examination of the parents showed no anomalies, demonstrating that the aberration occurred de novo in the fetus. Note, a genome-wide NIPT analysis on a blood sample taken at 27 weeks GA failed to detect this anomaly despite its size, indicating this to be a true fetal mosaicism of type 5 (TFM5).

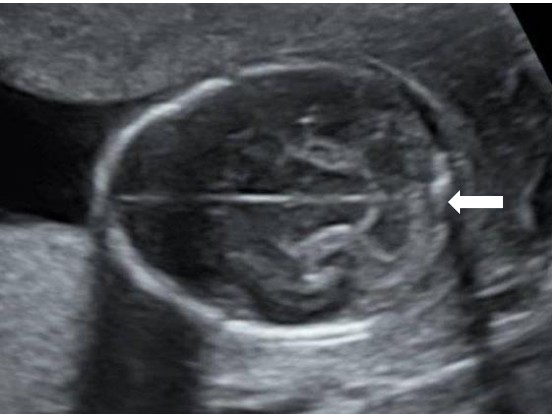

**Figure 1.** Subtle posterior curving of the cerebellum at 17 weeks.

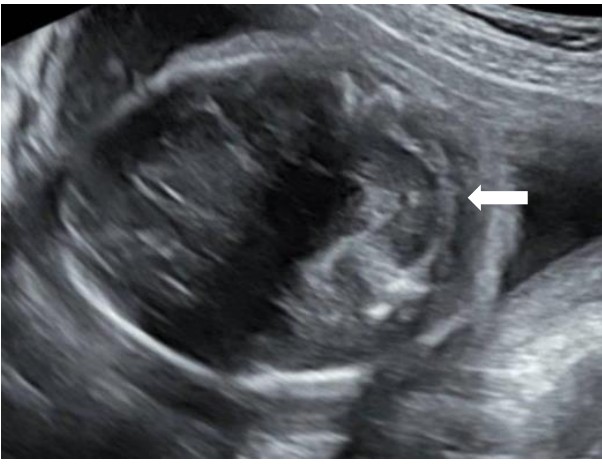

**Figure 2.** Overt banana-shaped cerebellum at 24 weeks.

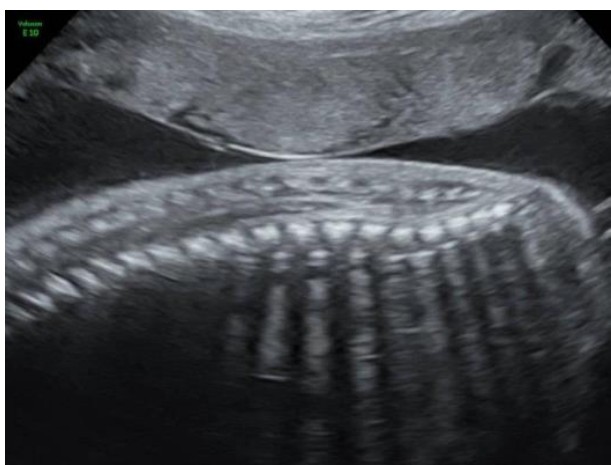

**Figure 3.** Inconspicuous aspect of the spine at 24 weeks.

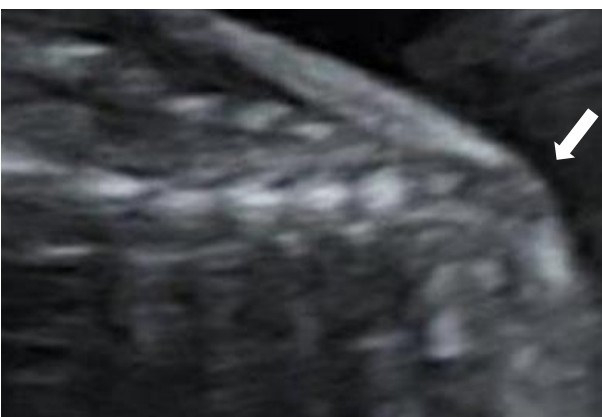

**Figure 4.** Minimal fistular connection between skin and spinal canal.

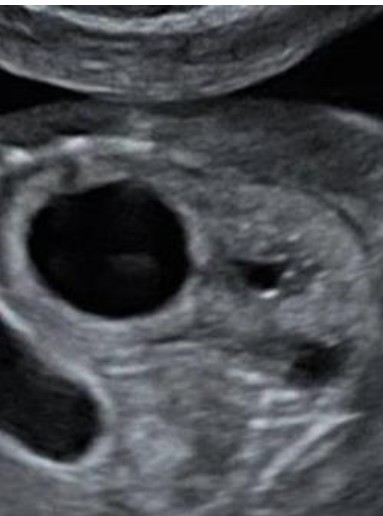

**Figure 5.** Hydronephrotic upper pole of the left kidney.

Duplication of the 3q region causes the 3q microduplication syndrome, which is associated with dysmorphic features, such as a low frontal hairline, excessive hair growth, and joined eyebrows as well as hand and foot abnormalities, defects of the internal organs (especially the heart), urinary tract anomalies, learning disabilities, and delayed development [1–4]. Similar phenotypes are described in patients with a mosaic marker chromosome or a triplication of this region, as diagnosed in this fetus. The causative gene(s) are thought to reside within the 3q26q27 segment [5].

Given the multiple congenital anomalies, the parents requested the termination of pregnancy (TOP) and, after seeking approval from the ethical committee of the university hospital, TOP was executed at 27 + 3 weeks GA. Postmortem investigation demonstrated and confirmed multiple malformations, including dysmorphic facial features, aberrant limbs with clinodactyly of the hands and syndactyly of the right foot, a sacral dimple (Figures 6 and 7) and a syringomyelia, a pulmonary valve stenosis with an overriding aorta, and hydroureter with hydronephrosis.

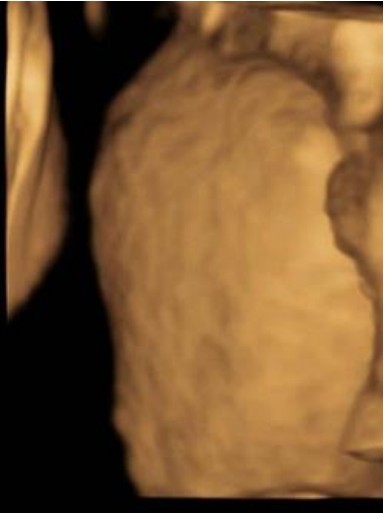

**Figure 6.** 3D ultrasound image of the lumbosacral region.

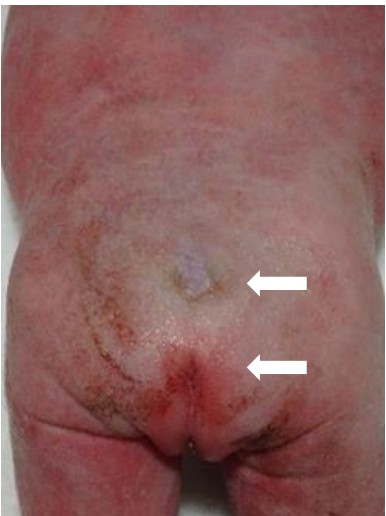

**Figure 7.** Postmortem sacral dimple (upper arrow) and opening of a sinus (lower arrow).

## 3. Discussion

Neural tube defects (NTD) occur in approximately 9/10,000 births in Europe, making them the most prevalent central nervous system anomalies. They encompass a whole spectrum of brain, spine, and spinal cord defects, including anencephaly, spina bifida, and encephalocele. NTDs can be categorised into open and closed types. Open defects include myelomeningocele, a congenital anomaly caused by incomplete closure of the distal neural tube during embryogenesis. In addition to the defect of the spine, open lesions are characterized by changes in brain morphology and morphometry, such as the typical banana and lemon-sign, ventriculomegaly, and hindbrain herniation. Examples of closed lesions are encephalocele, meningocele, and spina bifida occulta.

Here, we describe a case of a closed-type NTD with an unusual prenatal presentation of Arnold–Chiari malformation. Upon postmortem examination, dorsal dermal sinus was diagnosed. Dorsal dermal sinus is a subtype of spina bifida occulta. Different types of overlying skin lesions increase the suspicion of an occult spinal lesion. An abnormal hair tuft, a skin tag, an hemangioma, a subcutaneous lipomatous mass, a pigmented macule, or a dimple can be seen in the affected area in certain cases. Spina bifida occulta is usually characterised by minor abnormalities of the spine and can be accompanied by a tethered cord syndrome.

We performed a literature search for cases of prenatally diagnosed closed spina bifida lesions; however, they are rarely reported during prenatal ultrasound. Korsvik et al. described typical ultrasound findings and categorization of occult spina bifida [6]. They described dorsal dermal sinus but only upon postnatal imaging. Figuinha Milani et al. published a small series of prenatally diagnosed closed spina bifida cases that were characterized either by meningocele or by lipomas [7]. They emphasised that a thorough examination of the spine in sagittal, coronal, and transverse views is necessary to diagnose closed defects. This case demonstrates that, indeed, by using their technique with meticulous attention, it is possible to diagnose even the smallest defects. Similarly, Ghi et al. found in their series of spina bifida lesions that all prenatally detected closed lesions were characterised by a cystic mass [8]. The prenatal diagnosis of a closed spina bifida lesion is more difficult because secondary cranial findings, typically seen in open NTDs, are absent, and the spinal changes are more subtle [9]. The diagnoses can be made in subtle cases, such as this one, by identifying a lower position of the conus medullaris. On the other hand, it is a difficult examination that is not suitable for routine screening but can help if suspicion is raised. Dorsal dermal sinuses are even more difficult to diagnose, as there is no presence of a cyst or mass. In these cases, the nerves and spinal cord are not usually affected and, therefore, in contrast to open lesions, this disease has a minimal functional

impact. To our knowledge, this is the first case of closed spina bifida with a dorsal dermal sinus that is accompanied by secondary cranial abnormalities that has been diagnosed prenatally. In this fetus, a severe hindbrain herniation was detected while prenatal imaging demonstrated an intact spine and skin, both on MRI and ultrasound. Only by means of expert ultrasound, after increased suspicion of the diagnosis of spina bifida due to the elevated alfa 1-fetoprotein, could we find a small fistula between the skin and the spinal canal.

Following genetic analysis, the fetus was found to have a mosaic 3q23qter triplication that was present as a marker chromosome; this is the first case of this type. Previous reports have associated partial duplication of chromosome 3q with a broad phenotypic spectrum. Different studies have shown that a triplication in this region has a similar phenotype to the 3q duplication syndrome. Ounap reported a neonate with a triplication of the 3q25.3–q29 segment that had similar anomalies to our case, i.e., spina bifida, kidney, and limb anomalies [10].

## 4. Conclusions

Inspection of the brain is essential in the prenatal diagnosis of spina bifida. Most forms of open spina bifida are associated with Arnold–Chiari malformation. Closed forms of spina bifida are seldom associated with intracranial lesions and are, therefore, often missed by prenatal ultrasound even when an evaluation of the spine is performed in multiple planes. It would, therefore, be prudent not to make strong statements on the exclusion of occult spina bifida based on prenatal ultrasound; on the other hand, even the smallest defects can be detected by prenatal ultrasound if the spine is investigated carefully. When intracranial abnormalities are detected prenatally in a fetus with a seemingly normal aspect of the spine, evaluation of alfa-1-fetoprotein in the amniotic fluid and close examination of the skin overlying the spine can lead to the subtle diagnosis of dorsal dermal sinus.

**Author Contributions:** O.L. collected and studied the literature, contributed to idea development and wrote the manuscript. L.V.d.V. studied the literature and wrote part of the manuscript. B.B. wrote part of the manuscript and reviewed the different version. K.J. wrote part of the manuscript, and reviewed the final version. All authors have read and agreed to the published version of the manuscript.

**Funding:** This research received no external funding.

**Institutional Review Board Statement:** The study was conducted according to the guidelines of the Declaration of Helsinki, and approved by the Institutional Ethics Committee (University Hospital Antwerp, 2650 Edegem, Belgium). Protocol code: EC/PM/RVE/2021.056.

**Informed Consent Statement:** Written informed consent, including consent for publication, was obtained from the patient.

**Conflicts of Interest:** The authors declare no conflict of interest.

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
