# Peer review of "Pushing the Limits of Prenatal Ultrasound: A Case of Dorsal Dermal Sinus Associated with an Overt Arnold–Chiari Malformation and a 3q Duplication"

_2673-3897, doi:10.3390/reprodmed2030012_

Round 1

Reviewer 1 Report

Fig 4 could not be properly viewed in the manuscript and there was no mark to indicate de small fistula of the dermal sinus. 

Cardiac important anomalies are  described in the postmortem report but there are no information about prenatal evaluation of the heart and the overlooked findings should be mentioned.

Author Response

Dear Editor,

Please find below our responses to the reviewers comments. We hope the reviewers agree with our proposed changes.

Kind regards,

Olivier

Reviewer 1:

Fig 4 could not be properly viewed in the manuscript and there was no mark to indicate de small fistula of the dermal sinus.

Thank you for this comment. We adapted the figure.

Cardiac important anomalies are  described in the postmortem report but there are no information about prenatal evaluation of the heart and the overlooked findings should be mentioned.

Thank you for highlighting this shortcoming. We added the cardiac ultrasound findings to the case description:
“Cardiac sonography was normal both at 17 and 24 weeks GA, with normal four-chamber view, outflow tracts and three-vessel view.”

Reviewer 2 Report

Dear authors, I have reviewed your manuscript entitled “Pushing the limits of prenatal ultrasound: a case of dorsal dermal sinus associated with an overt Arnold-Chiari malformation and a 3q duplication“. Overall this case report is relatively interesting. The topic is relevant, so it may be of great interest to the readers. However, the manuscript should be further improved.

Major revision will need to be performed, however, to address the following comments. In general, the authors should clarify the following points:

  • Introduction section should introduce the reader with general concept of the manuscript. The section introduction is too short.
  • I recommend you to clearly define the aim of your article. This info is necessary to by specified in the section introduction.
  • Please reorganize the figures. You cited in the text figure 1 and the second figures is no 5. The figures should appear in the text in order 1,2,3,4,5…..
  • It may be of great interest a short review of the literature, in order to help readers.
  • Knowing these results, what are the conclusions with regard to improving care?
  • The section references needs some minor revision in accordance with Instructions for Authors ( Author 1, A.B.; Author 2, C.D. Title of the article. Abbreviated Journal NameYearVolume, page range.)
  • Generally, the manuscript needs some minor punctuations end English editing.

Author Response

Dear Editor,

Please find below our responses to the reviewers comments. We hope the reviewers agree with our proposed changes.

Kind regards,

Olivier

Reviewer 2:

Dear authors, I have reviewed your manuscript entitled “Pushing the limits of prenatal ultrasound: a case of dorsal dermal sinus associated with an overt Arnold-Chiari malformation and a 3q duplication“. Overall this case report is relatively interesting. The topic is relevant, so it may be of great interest to the readers. However, the manuscript should be further improved.

Major revision will need to be performed, however, to address the following comments. In general, the authors should clarify the following points:

Introduction section should introduce the reader with general concept of the manuscript. The section introduction is too short.

Thank you for this comment. We agree that the introduction was short, we adapted this. The text now reads:
“Neural tube defects are the most common central nervous system anomaly. Two major types are distinguished: the open spina bifida aperta and closed spina bifida occulta. Open spina bifida lesions are usually diagnosed prenatally by ultrasound based on the characteristic brain abnormalities: the ‘lemon-sign’ which reflects the abnormal skull shape and the ‘banana-sign’ reflecting the abnormal cerebellar shape. Furthermore, in open spina bifida, a dorsal sac (myelomeningocele) or missing processi spinosi can easily be seen on ultrasound which confirm the diagnosis. Closed spina bifida lesions are usually more difficult to diagnose on prenatal ultrasound because of the absence of cranial findings. If the lesion is characterized by a skin covered sac (meningocele), the diagnosis can still be made. However certain subtypes of closed spina bifida are more subtle and are therefore often missed on prenatal ultrasound. Dorsal dermal sinuses are one of the subtypes of closed spina bifida lesions. These lesions are characterized by an epithelial-lined tract that connects the skin surface with the intracanalicular space. Because of the absence of secondary cranial abnormalities, as found in open spina bifida lesions, and the absence of cysts or masses at the spine, these lesions are usually missed on prenatal ultrasound screening. Alfa-fetoprotein levels can be measured in amniotic fluid to help finding the diagnosis. If there is leakage of spinal fluid, these levels will rise. This technique is nowadays only rarely performed, however in difficult or atypical cases it can still help to confirm the diagnosis. We present a case of a fetus with cranial abnormalities typical of open spina bifida but with an intact spine on ultrasound and fetal MRI.”

I recommend you to clearly define the aim of your article. This info is necessary to by specified in the section introduction.

Thank you for this comment. We added this to the introduction:

“Herein we want to demonstrate that even in closed spina bifida, cranial abnormalities can be present.  Furthermore, when cranial abnormalities are detected on prenatal ultrasound but the spine seems intact, alfa-foetoprotein remains a powerful tool to help diagnose prenatal atypical spina bifida cases..”

Please reorganize the figures. You cited in the text figure 1 and the second figures is no 5. The figures should appear in the text in order 1,2,3,4,5…..

Thank you for noticing this. We adapted the figures.

It may be of great interest a short review of the literature, in order to help readers.

Thank you for this suggestion. We performed a review of literature and extended this part of the discussion.

We performed a literature search for cases of prenatally diagnosed closed spina bifida lesions, however they are rarely reported on prenatal ultrasound. Korsvik et al described typical ultrasound findings and categorization of occult spina bifida (6). They described dorsal dermal sinus but only on postnatal imaging. Figuinha Milani et al published a small series of prenatally diagnosed closed spina bifida cases which were characterized either by meningocele or by lipomas (7). They emphasized that a thorough examination of the spine in sagittal, coronal, and transverse views is necessary to diagnose closed defects. This case demonstrates that indeed, using their technique, with meticulous attention, it is possible to diagnose even the smallest defects. Similarly, Ghi et al found in their series of spina bifida lesions that all prenatally detected closed lesions were characterized by a cystic mass (8). The prenatal diagnosis of a closed spina bifida lesion is harder because secondary cranial findings, typically seen in open NTDs, are absent and the spinal changes are more subtle (9). The diagnoses can be made in subtle cases such as this one, by identifying a lower position of the conus medullaris. That on the other hand is a difficult examination not suitable for screening but can help if suspicious is raised. 

Knowing these results, what are the conclusions with regard to improving care?

Thank you for this question. We clarified this in the conclusion that closed spina bifida is a difficult diagnosis however even when the spine seems intact meticulous evaluation of the spine can demonstrate very small defects. Secondly although alpha-fetoprotein is only rarely used to diagnose spina bifida, it is still a powerful tool in atypical cases. Thirdly, until now it was generally accepted that closed spina bifida was linked to cranial abnormalities but this case demonstrates that this is (rare but) possible. Lastly, this case ads to the knowledge on the phenotype of 3q24-qter triplication.

The section references needs some minor revision in accordance with Instructions for Authors ( Author 1, A.B.; Author 2, C.D. Title of the article. Abbreviated Journal NameYear, Volume, page range.)

We apologies for this error. We adapted the reference list with endnote.

Reviewer 3 Report

This is a "small-print" but nevertheless interesting observation, satisfactorily documented and well illustrated, and worth publishing. The English could be a little more elegant, but it is perfectly comprehensible.

One small point: could the authors note which Genome Build is used for cytogenetic coordinates of the chromosomal imbalance. And I'd suggest the title say 3q24q27, not just 3q.

Author Response

Hello, 

Thank you for your comment.  I submitted our paper for English editing to our editing service at the Antwerp university.  The responsable lady is a native Englishwoman...

I corrected the paper following her instructions.

The Genome Build used is : GRCh37

Since The 3q duplication is an known entitiy, we would like to keep the title as such.

Kind regards,

Olivier

Reviewer 4 Report

Comments/questions

Line 63: Was this cardiac screening performed at the time of the anatomy scan or were these fetal echocardiograms? Do you have an explanation for how the pulmonic stenosis and overriding aorta were missed?

Page 3/4: Looks like images/photos included twice

Lines 164-169: Is your case the first mosaic 3q24-qter triplication reported with closed spina bifida? Just wasn't clear. 

Line 180: You conclude that amniotic fluid alpha-fetoprotein may be useful -- and it seems reasonable to send off once one is performing an amniocentesis for genetic studies, but do you feel you would not have identified the defect without this information?

Overall, a nice case report that adds to the literature. 

Author Response

Hello, 

Thank you for your comment.  I submitted our paper for English editing to our editing service at the Antwerp university.  The responsable lady is a native Englishwoman...

I corrected the paper following her instructions.

For your intrests : line 63 the heart anomalies were subtile on pathology.  Sometimes we feel that the pathological diagnosis is black of white, but in this case is was a rather discrete defect.

line 164 : It is indeed the first genetic anomaly of its kind. I noted this in the paper.

line 180 : We suspected the anomaly on ultrasound.  There had to be a leakage of CSF somewhere.  The AFP levels just proved it.

Kind regards,

Olivier

Round 2

Reviewer 2 Report

Dear authors,

I agree with your changes.

The manuscript needs some minor punctuations end English editing.

Author Response

Hello, 

Thank you for your comment.  I submitted our paper for English editing to our editing service at the Antwerp university.  The responsable lady is a native Englishwoman...

I corrected the paper following her instructions.

Kind regards,

Olivier
